# Effects of Different Factors on Single Event Effects Introduced by Heavy Ions in SiGe Heterojunction Bipolar Transistor: A TCAD Simulation

Zheng Zhang [1] , Gang Guo [1,*], Futang Li [1], Haohan Sun [1], Qiming Chen [1], Shuyong Zhao [1], Jiancheng Liu [1] and Xiaoping Ouyang [1,2]

[1] Department of Nuclear Physics, China Institute of Atomic Energy, Beijing 102413, China
[2] Northwest Institute of Nuclear Technology, Xi'an 710024, China
* Correspondence: ggg@ciae.ac.cn

**Abstract:** In this paper, the effects of different factors, including the heavy ions striking location, incident angle, linear energy transfer (LET) value, projected range, ambient temperature and bias state, on the single event transient introduced by heavy ions irradiation in the SiGe heterojunction bipolar transistor (HBT) were investigated by the TCAD simulation. The results show that the current transient peak value, collected charge and carrier type of each terminal are changed by the striking location, incident angle and bias state. The current transient peak value and collected charge increase with the LET value, while they decrease with the ambient temperature. When heavy ions vertically irradiate the collector and substrate, the current transient peak value and collected charge increase with the projected range; therefore, the species of heavy ions should be considered in studying the single event effects of the SiGe HBT induced by heavy ions irradiation. The microphysical mechanism of these factors influencing the single event effects of the SiGe HBT is discussed in this work.

**Keywords:** SiGe heterojunction bipolar transistor; single event effect; single event transient; charge collection; TCAD simulation

## 1. Introduction

The space radiation environment is filled with a large number of high-energy charged particles, which will inevitably affect the electronic components of space missions [1–3]. The high-energy charged particles mainly come from galactic cosmic rays, solar cosmic rays and the Van Allen radiation belt, and they deposit energy into the aerospace devices to cause radiation effects, resulting in the functional failure of the aerospace devices and even the failure of the related space missions. The radiation effects of the aerospace devices can be divided into the ionizing effect and the non-ionizing effect. The ionizing effect mainly includes single event effects [4–6] and the total ionizing dose effect [7–9], while the non-ionizing effect is mainly the displacement damage effect [10,11]. These radiation effects occur simultaneously in the aerospace devices and interact with each other [12]. Since the launch of the first man-made satellite, 46% of spacecrafts and satellites suffered functional failures due to these radiation effects [13], ultimately resulting in mission failure.

In addition to the radiation effects caused by high-energy charged particles in the space radiation environment, the aerospace devices also face the challenge of an extremely low-temperature environment during the space mission. There are a lot of extremely low temperatures in the space environment, for example, the temperature on the Mars surface usually ranges from −133 °C to 27 °C, the temperature range of the lunar rover during its mission is usually from −180 °C to 120 °C and the lowest temperature at the polar craters on the lunar surface reach −230 °C. Therefore, it is of great significance to develop an aerospace device with excellent radiation resistance and extremely low-temperature characteristics for the future aerospace industry. Once the research is successful, a large amount of thermal

insulation equipment can be removed, which would not only reduce the cost of launching spacecraft and satellites but also enhance their deep-space exploration capabilities.

Since the early 2000s, NASA has been concerned about the use of electronic systems in the extreme space environments. Previous studies have indicated that the germanium silicon heterojunction bipolar transistor (SiGe HBT) has excellent total dose radiation resistance [14,15] and excellent low-temperature characteristics [16,17] due to the advantages of the silicon-based energy band engineering materials, semiconductor process and device structures. The SiGe HBT can operate normally in the temperature range of $-180\ ^{\circ}$C to 125 $^{\circ}$C, with the total ionizing dose effect resistance up to an Mrad(Si) magnitude [18,19] and the displacement damage resistance up to a $10^{15}\ \mathrm{cm}^{-2}$ magnitude (equivalent fluence of 1 MeV neutron) [20]. Therefore, the SiGe HBT has an attractive application prospect in the field of the extreme space environment. However, a large number of studies have found that the SiGe HBT is very sensitive to the single event effects [21,22] and has a complex charge collection mechanism different from traditional bulk silicon devices. Thus, the study of the single event effects has always been a hot topic in the research of SiGe HBT radiation effects.

In recent years, many scholars have conducted a significant amount of research on the single event effects of the SiGe HBT. Many famous research institutions such as the Georgia Institute of Technology, Auburn University, Vanderbilt University, and the Boeing company have carried out a lot of research works on the single event effects of the commercial SiGe HBT produced by IMB, National Semiconductor, Jazz Semiconductor and other companies [23], in which a lot of the research works have been carried out based on the fourth-generation SiGe HBT produced by the IBM company. Since 2005, the Georgia Institute of Technology and Auburn University have studied the charge collection mechanism, key influencing factors and anti-radiation reinforcement design of the SiGe HBT single event effects with the help of proton, heavy ion and laser microbeam irradiation experiments and the TCAD numerical simulation [24,25]. The results show that the sensitive area of single event effects in the SiGe HBT with a deep trench isolation (DTI) structure is the DTI region. The DTI structure can not only prevent the external excess carriers from diffusing to the inner collection node but also limit the excess carriers from diffusing to the outside, resulting in a significant increase in the charge collection. The transient current amplitude and integral charge collection induced by the single event effects are closely related to the linear energy transfer (LET) value of the incident ions, but also to the projected range of the incident ions, indicating that the light-doping substrate in the bulk silicon process has an important effect on the sensitivity of the SiGe HBT single event effects. In 2015, Li et al. used laser microbeams to study the single event effects of the SiGe HBT produced by a Chinese company and IBM, respectively [26,27]. Due to the similar doping concentration in their collector region, the peak current values of the two SiGe HBTs collectors were close. The SiGe HBT produced by the Chinese company has a large C/S junction, and its collector has a strong charge collection capacity, so the transient current pulse width is large. From 2017 to 2019, Wei et al. studied the single event effects of the SiGe HBT produced by the Chinese company through the heavy ion microbeam and proton irradiation experiments [28]. The results show that the collector transient current peak value caused by heavy ions is significantly higher than that caused by a proton under the same bias state. The transient current pulse width of the SiGe HBT collector caused by heavy ions is also wider than that caused by protons. When the SiGe circuit works at a higher frequency and its working period is shortened to a time scale of ps, the collector transient current pulse induced by heavy ions covers more working periods than that of the collector transient pulse induced by protons, thus inducing a more serious multi-bit upset effect.

Through a TCAD simulation, Zhang et al. found that the incident angle of heavy ions would significantly change the ionization track length in the SiGe HBT [29], which would lead to the difference in the charge deposition and ultimately the difference in the charge collection. In the SiGe HBT, the emission direction of secondary particles produced by

the intermediate and high-energy proton through a nuclear reaction is relatively random. Changing the incident angle of the proton can not uniquely determine the direction of the secondary particles; therefore, the single event effects in the SiGe HBT can not be changed significantly by changing the incident angle. Through a Monte Carlo simulation, Wei et al. found that when the SiGe HBT was irradiated by the proton in different incident angles [30], the main body of the collector transient current pulse waveform distribution shows the same basic characteristics, a fast rising edge and a relatively slow falling edge. However, as the incident angle of the proton increases, the falling edge becomes very slow, and the distribution range of the collector transient current pulse duration expands.

Above all, the single event effects in the SiGe HBT can be well reproduced by a TCAD simulation, which is not affected by the running time compared with the experimental study. At the same time, a TCAD simulation can complete the research content that is difficult to achieve in the experiment, so as to provide a theoretical basis for the practical application of the SiGe HBT in a space environment. Based on the process and structure of the SiGe HBT produced by the Chinese company, the effects of the striking location, incident angle and LET value of heavy ions, ambient temperature, bias state and other factors on the single event effects of the SiGe HBT were carried out by the TCAD simulation in this paper. The sensitive area of the charge collection, the effects of these factors on the current transient pulse peak value and width of each terminal and the charge collection amount were determined, which provides further theoretical support for the radiation-hardening technique of the SiGe HBT produced by the Chinese company.

## 2. Materials and Methods

In this paper, a domestic SiGe HBT is selected as the research object, whose inner structure is similar to that of the traditional bulk silicon npn vertical bipolar transistor, as shown in Figure 1. The base region is composed of SiGe material with gradual change in components. The introduction of Ge in the base region forms a slow mutation heterojunction at the emitter/base pole junction (E/B junction) and base/collector junction (B/C junction), as shown in Figure 2. The built-in electric field formed in the base region effectively improves the carrier transit time in the base region. Current gain increases exponentially with the band-gap variation $\triangle E_g$ after the introduction of Ge, as shown in the following equation:

$$h_{fe} = \frac{N_e}{N_b} \frac{V_{nb}}{V_{pe}} exp(\triangle E_g / kT) \tag{1}$$

where $h_{fe}$ is the current gain, $N_e$ is concentration in emitter region, $N_b$ is the concentration in base region, $V_{nb}$ is the electrons' velocity in base region, $V_{pe}$ is the holes' velocity in emitter region, $\triangle E_g$ is the band-gap change, $k$ is the Boltzmann's constant and $T$ is the temperature. The base region thickness of the SiGe HBT is 0.08 μm, and the doping concentration is up to $10^{19}$ cm$^{-3}$, which effectively reduces the resistance of the base region and enables the SiGe HBT to simultaneously meet the requirements of high frequency and high gain. A shallow trough isolation (STI) was used to form an active region in the region from the base to the collector. Above the isolated oxide layer, a polysilicon layer doped boron and germanium exported the base, which was epitaxial by the dual polysilicon self-alignment process. Heavy doping epitaxial region can reduce resistance of the base region and the B/C junction. n$^+$ buried layer leads to the collector. The emitter region is manufactured using polysilicon and leads to the emitter contact at the top. Near the edge of the SiGe HBT, boron ions are injected by ion implantation process to form a P-type isolation wall and export the substrate.

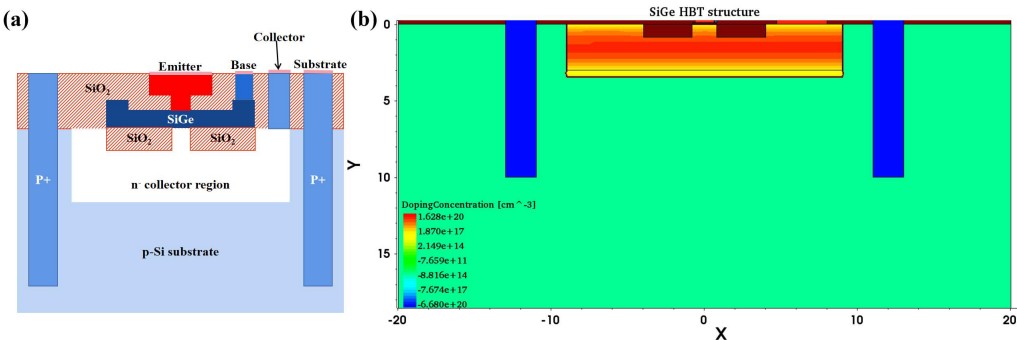

**Figure 1.** Schematic diagram (**a**) and the internal structure simulation profile (**b**) of the SiGe HBT.

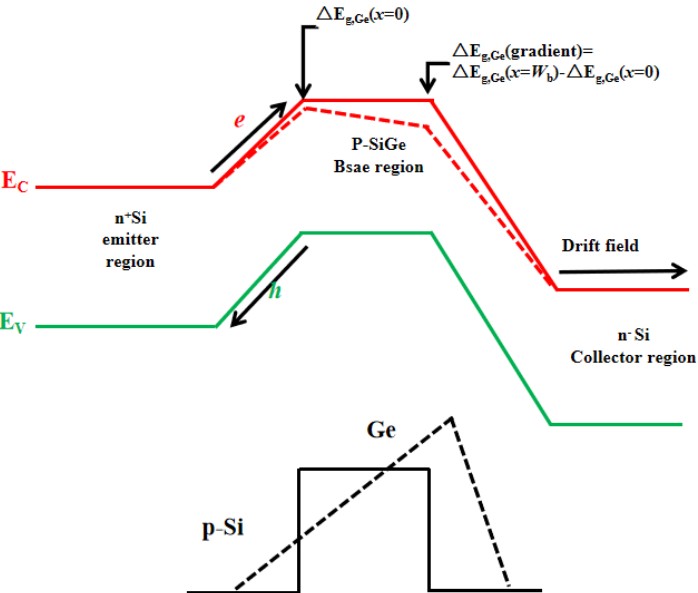

**Figure 2.** Schematic diagram of the band structure of the SiGe HBT.

Philips unified mobility model, SRH recombination model, Auger recombination model, velocity saturation model and band-gap narrowing model were used as physical models in TCAD simulation. The majority and minority carrier mobility for bipolar transistor can be accurately simulated by the Philips unified mobility model. The high concentrations of electron and hole in the SiGe HBT can be described by the SRH and Auger recombination models. The velocity saturation model is used due to the presence of high carrier density gradient. The band-gap narrowing model is used because germanium doping will cause gradual change in the band structure.

When heavy ions are striking on the SiGe HBT, a large number of electron–hole pairs are generated by ionization along the ions track, which distorts the potential in the depletion layer and forms a funnel potential toward the substrate. Under the action of the funnel electric field and concentration gradient, carriers are rapidly collected by each terminal through drift and diffusion, and such a large amount of charge collection will cause changes in the current of each terminal in a short time (a few nanoseconds). In TCAD simulation, the calculation of carrier generation rate caused by heavy ions irradiation is the key. The number of electron–hole pairs before the initial heavy ions striking is added to the carrier density at the beginning of the simulation, and the carrier generation rate after heavy ion incident is given by the following formula:

$$G(l, \omega, t) = G_{LET}(l) R(\omega, l) T(t) \tag{2}$$

where $R(\omega, l)$ and $T(t)$ represent the carrier generation rate as a function of space and time, respectively. Changing carrier with time is Gaussian distribution, that is, $T(t)$ can be expressed by Equation (3):

$$T(t) = \frac{2 \cdot exp(-(\frac{t-t_0}{\sqrt{2} \cdot S_{hi}})^2)}{\sqrt{2} \cdot S_{hi} \sqrt{\pi}(1 + erf(\frac{t_0}{\sqrt{2} \cdot S_{hi}}))} \tag{3}$$

where $t_0$ is the moment when heavy ions enter the SiGe HBT, $S_{hi}$ is the Gaussian characteristic value. Changing carrier with space can follow either exponential function or Gaussian function. Gaussian distribution is used in this paper, and Equation (4) is a function representation of $R(\omega, l)$.

$$R(w, l) = exp(-(\frac{\omega}{\omega_t(l)})^2) \tag{4}$$

where $\omega$ is the vertical distance to the ion track, and $\omega_t(l)$ is the characteristic length. The striking locations of heavy ions on the SiGe HBT are, respectively, set in the center of emitter, base, collector and substrate.

Figure 3 shows the Gummel characteristic curve of the SiGe HBT obtained by TCAD simulation in this paper; it is in good agreement with the simulation results obtained by others and the tested values by the semiconductor parameter tester KETHLEY4200. This indicates that the SiGe HBT structure model constructed in this work can accurately reflect the actual performance of the SiGe HBT. In this paper, the substrate is biased at −5 V and all other terminals are biased at 0 V to achieve the worst bias, except where it is specifically stated, for example, in the study of the effects of bias state on the single event effects of the SiGe HBT induced by heavy ions.

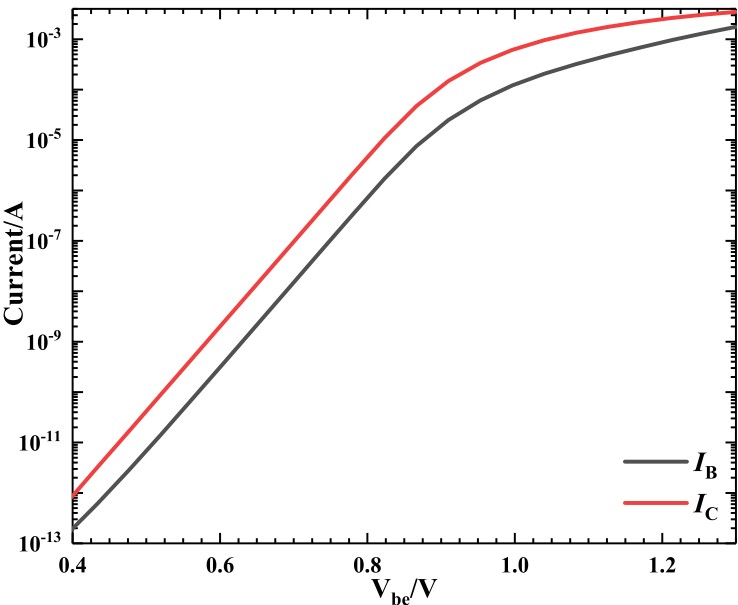

**Figure 3.** Gummel characteristic curve of the SiGe HBT.

## 3. Results and Discussion

### 3.1. Striking Location

The single event effects' sensitive area of the SiGe HBT can be obtained by analyzing the device structure and the simulation results of heavy ions striking at different locations. The charge collection quantity at each terminal is closely related to the striking location of the heavy ions. Figures 4 and 5 show the current transients and charge collection quantity of each terminal introduced by heavy ions striking at the center of the emitter, base, collector and substrate, respectively. When heavy ions are striking at the center of the emitter and

base, electrons are collected by the emitter and collector, holes are collected by the base and the current of the substrate is not changed significantly by the heavy ions striking. When heavy ions are striking at the center of the collector and substrate, the electrons are collected by the collector, the holes are collected by the base and substrate and the current of the base and emitter is not changed significantly. The current transient peak value of each terminal is strongly dependent on the heavy ions' striking location. The current transient peak value caused by the irradiation of heavy ions on the emitter and base is much higher than that caused by the irradiation of heavy ions on the collector and substrate. When heavy ions are irradiating the collector, the current transient peak value is the minimum, which indicates that the emitter and base are the sensitive area of the single event transient. Electron–hole pairs are generated by ionization when the heavy ions are incident on the sensitive area of the SiGe HBT, the electrons are collected at the high potential region and the holes flow in the direction of decreasing potential. When heavy ions are striking the center of different terminals, different carrier transport modes result in the different response of the single event transient. When heavy ions irradiate the emitter and base of the SiGe HBT, the total charge quantity collected by the emitter and collector is equal to the charge quantity collected by the base. While heavy ions irradiate the collector and substrate of the SiGe HBT, the charge collected by the collector is equal to the charge quantity collected by the substrate. All of the results show that the emitter is the sensitive area of the SiGe HBT.

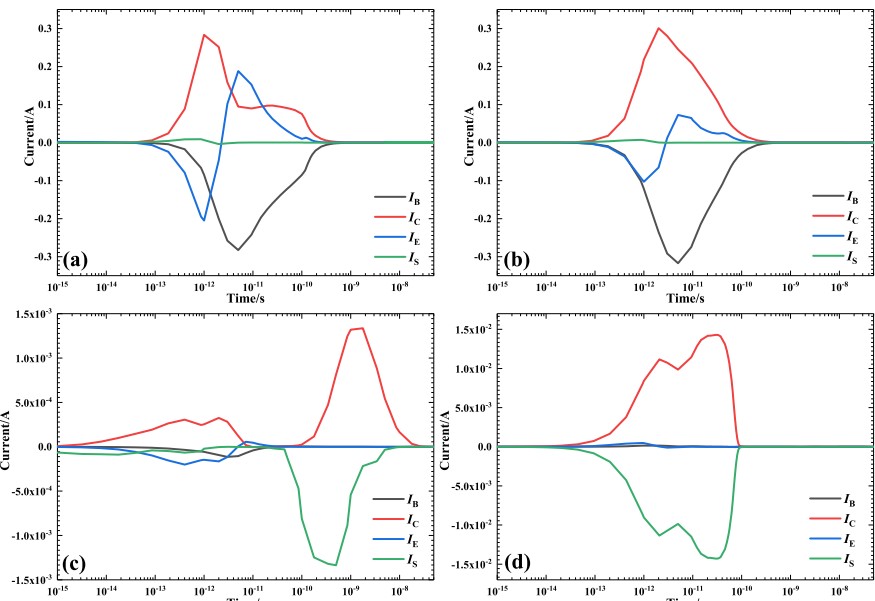

**Figure 4.** The current change in each terminal with time as heavy ions irradiate the center of emitter (**a**), base (**b**), collector (**c**) and substrate (**d**) of the SiGe HBT.

### 3.2. Incident Angle

The effective LET value of the heavy ions incident on the SiGe HBT surface changes with the incident angle. Figure 6 shows the current transients and collected charge of the base as the heavy ions irradiate the emitter and base of the SiGe HBT with different incident angles. With the increase in the incident angle, the current transient peak value increases first and then decreases, and the current transient pulse width does not change obviously. The collected charge quantity of the base changed with the incident angle, the collected charge quantity of the base is the maximum when the heavy ions irradiate the emitter with an angle of 60°, while the collected charge quantity of the base is the maximum when the heavy ions irradiate the base with an angle of 90°. As shown in Figure 1, the incident angle increases in a clockwise direction. The base current changes with the incident angle caused by the change in the distance from the carrier to the base, and it decreases first and then increases with the incident angle when the heavy ions irradiate the emitter of the SiGe HBT.

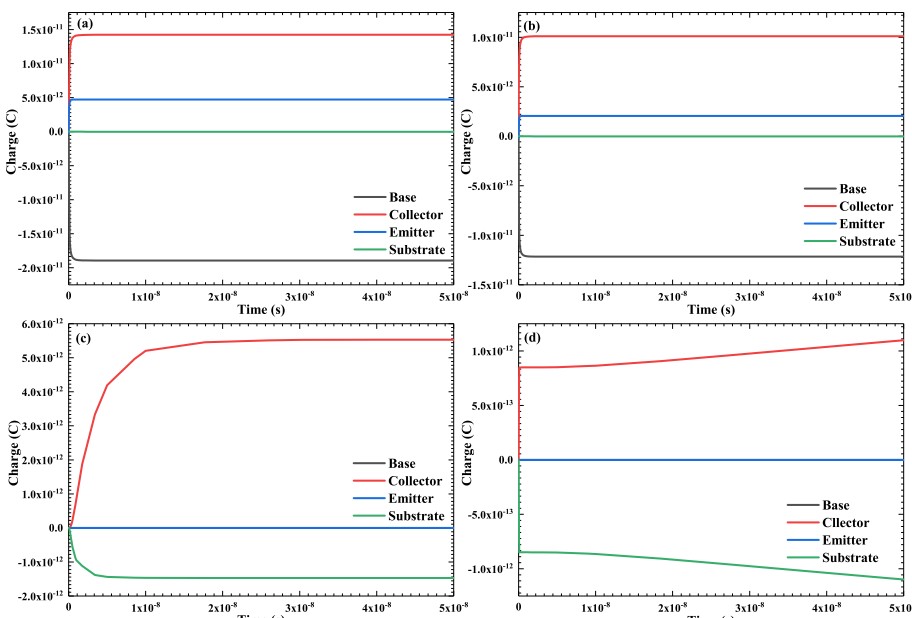

**Figure 5.** The collected charge change in each terminal with time as heavy ions irradiates the center of emitter (**a**), base (**b**), collector (**c**) and substrate (**d**) of the SiGe HBT.

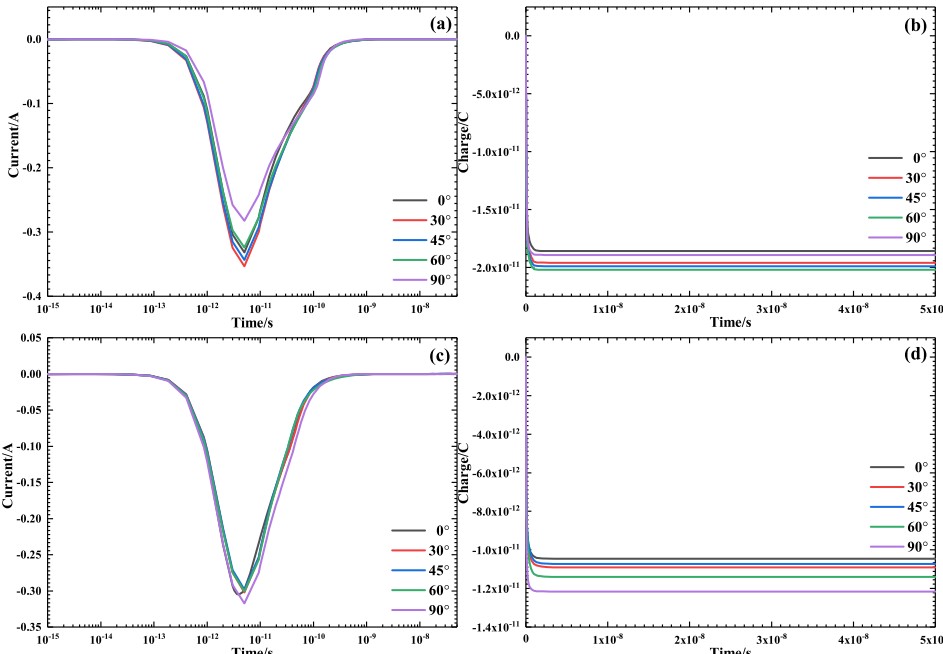

**Figure 6.** The current and collected charge change in the base with time as heavy ions irradiates the center of emitter (**a**,**b**) and base (**c**,**d**) of the SiGe HBT.

Figure 7 shows the current transients pulse and collected charge quantity of the collector as the heavy ions irradiate the collector and substrate of the SiGe HBT with different incident angles. When the heavy ions irradiate the collector, there are two transient peaks in the current of the collector caused by the drift and diffusion of electrons. The current transient peak value and collected charge quantity change with the incident angle. When the heavy ions irradiate the substrate, the current transient peak and pulse width of the collector are changed by the incident angle of the heavy ions. The current transient peak value and collected charge quantity of the collector caused by heavy ions irradiation with an angle of 0° is the lowest, while the values caused by heavy ions irradiation with an angle of 30° is the highest.

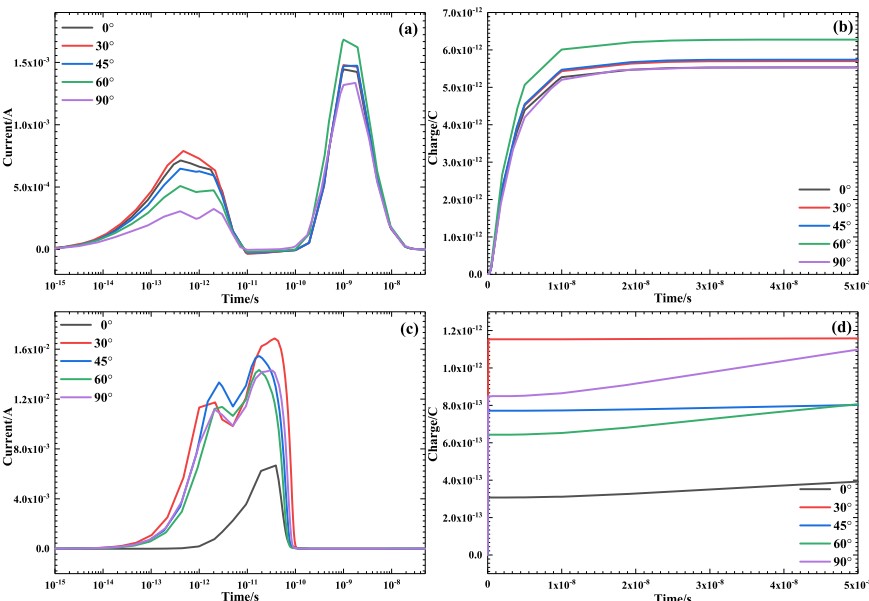

**Figure 7.** The current and collected charge change in the collector with time as heavy ions irradiate the center of collector (**a**,**b**) and substrate (**c**,**d**) of the SiGe HBT.

### 3.3. LET Values

The number of electron–hole pairs in the SiGe HBT produced by heavy ions irradiation with different LET values is different, and the number of the electrons and holes in SiGe HBT increases with the heavy ions' LET value. In the TCAD simulation, the LET value of the heavy ions is represented by the charge deposition quantity, and the charge deposition quantity of 0.1 pC/µm corresponds to the LET value of the heavy ions of 10 MeV·cm$^2$/mg. When studying the effect of the heavy ions' LET values on the single event effects of the SiGe HBT, the charge quantity deposited by the heavy ions in the SiGe HBT is set to vary in the range 0.1 to 1.5 pC/µm, and the corresponding heavy ion LET values vary in the range 10 to 150 MeV·cm$^2$/mg. Figures 8 and 9 show the change in the current and collected charge with time when the heavy ions with different LET values vertically irradiate the different terminals of the SiGe HBT. The current transient peak value, pulse width and collected charge increase with the heavy ions' LET values.

### 3.4. Projected Range

Different heavy ions with the same LET value have different projected ranges in the SiGe HBT, and lighter ions will have a longer projected range in the SiGe HBT. Whether the LET value can be used to characterize the single event effects induced by different heavy ions irradiation depends on the microstructure and the inner material of the device. Figures 10 and 11 show the current transients and collected charge quantity of the base and collector under the irradiation of heavy ions with the same LET value and different projected ranges, respectively. The projected range of heavy ions is set in the range of 1 to 10 µm. When heavy ions irradiate the emitter, the base current peak value and collected charge quantity increase with the projected range. When heavy ions irradiate the base, the base current peak value decreases with the projected range, but the collected charge is increased by the projected range because the transient pulse width increases with the projected range. When the heavy ions irradiate the collector and substrate, the current peak value, pulse width and collected charge quantity of the collector increase with the heavy ions' projected range. Therefore, the species of heavy ions should be considered when studying the single event effects of the SiGe HBT induced by heavy ions irradiation.

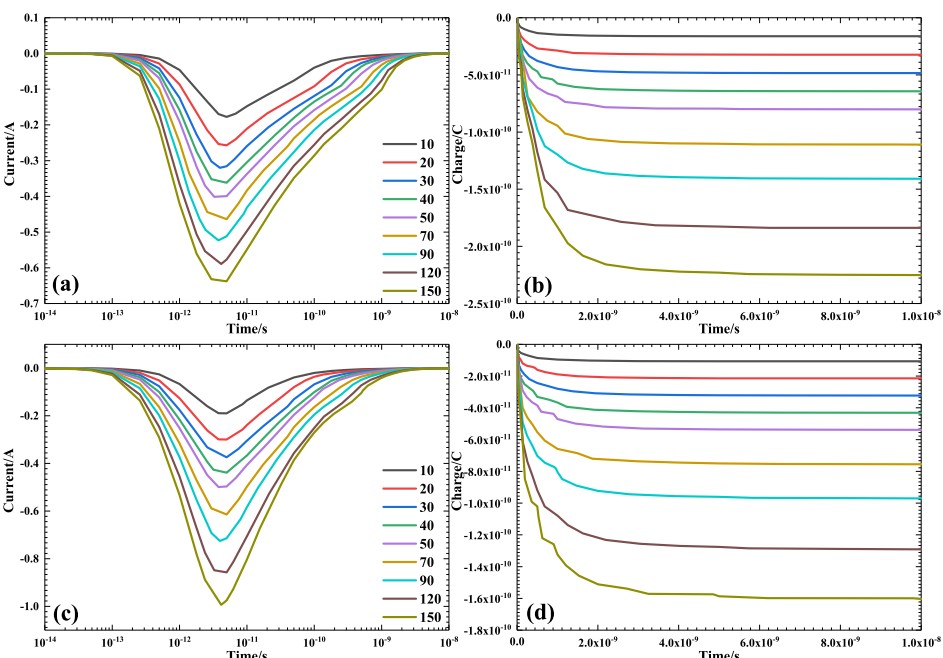

**Figure 8.** Change in base current and collected charge with time when heavy ions with different LET values vertically irradiate the emitter (**a**,**b**) and base (**c**,**d**) of the SiGe HBT.

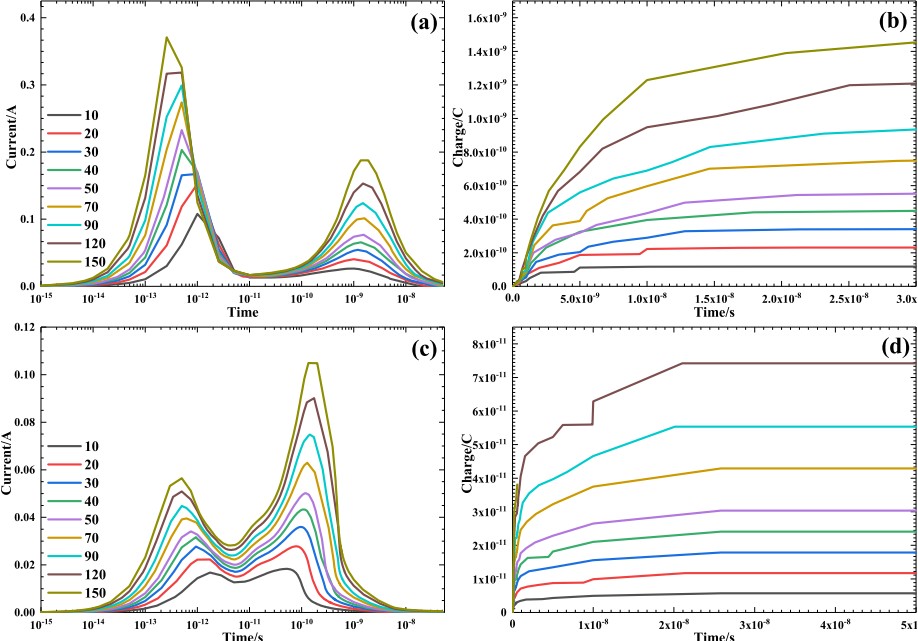

**Figure 9.** Change in collector current and collected charge with time when heavy ions with different LET values vertically irradiate the collector (**a**,**b**) and substrate (**c**,**d**) of the SiGe HBT.

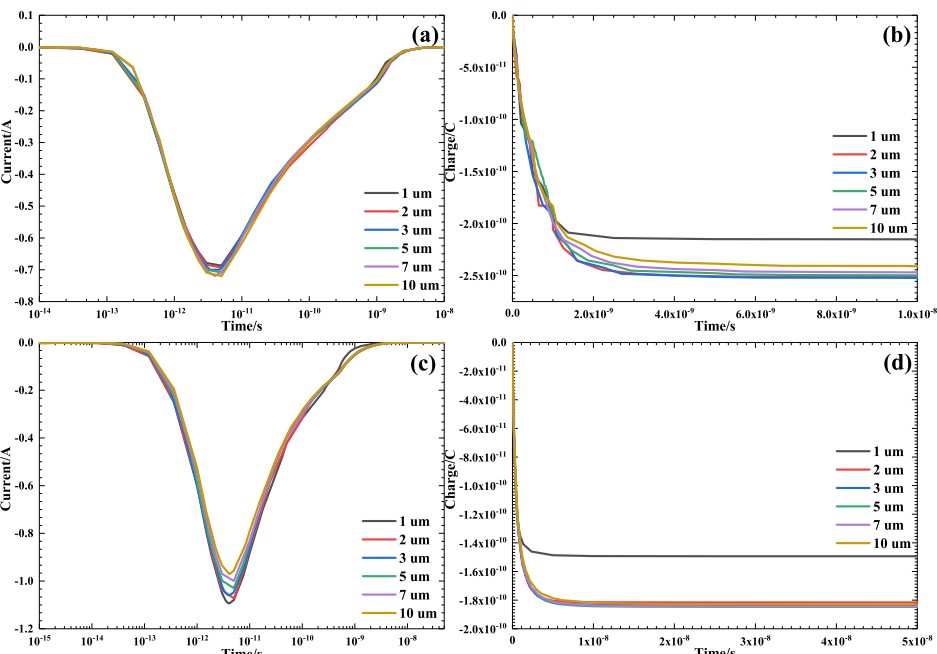

**Figure 10.** Change in base current and collected charge with time when heavy ions with different projected range vertically irradiate the emitter (**a**,**b**) and base (**c**,**d**) of the SiGe HBT.

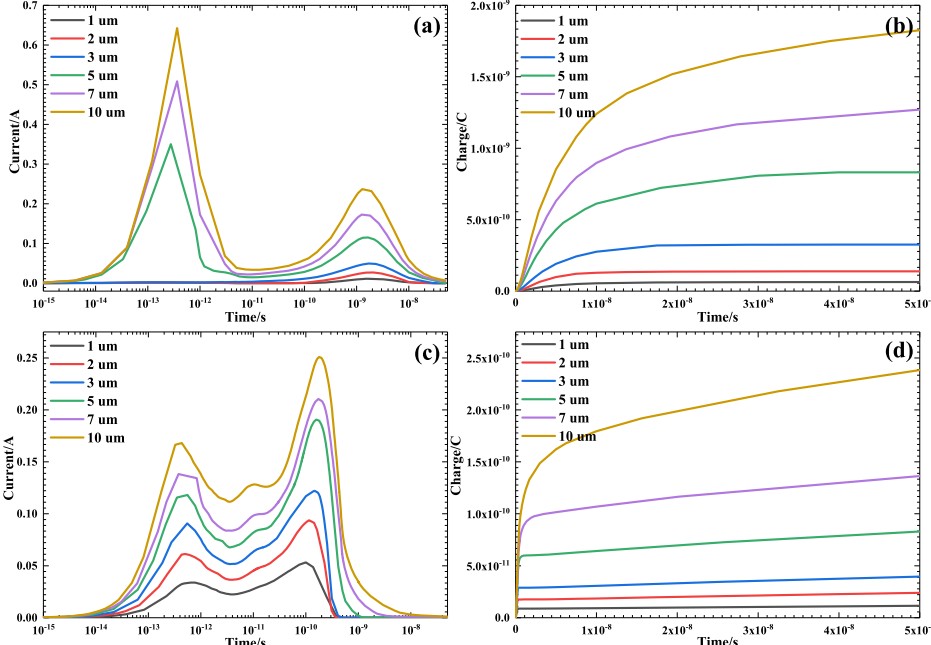

**Figure 11.** Change in collector current and collected charge with time when heavy ions with different projected range vertically irradiate the collector (**a**,**b**) and substrate (**c**,**d**) of the SiGe HBT.

### 3.5. Ambient Temperature

Under different temperatures, the mobility of the carrier in the SiGe HBT is different, and the mobility of the carrier decreases with the temperature. Therefore, the ambient temperature during the heavy ions irradiation has a significant influence on the single event effects of the SiGe HBT. Figures 12 and 13 show the current transients and collected charge quantity of the base and collector introduced by the heavy ions irradiation under different ambient temperatures. The current transient peak value and collected charge quantity decrease with the ambient temperature, while the pulse width is not significantly changed by the ambient temperature. We can conclude that increasing the ambient temper-

ature of the SiGe HBT can effectively reduce the single event effects caused by the heavy ions irradiation.

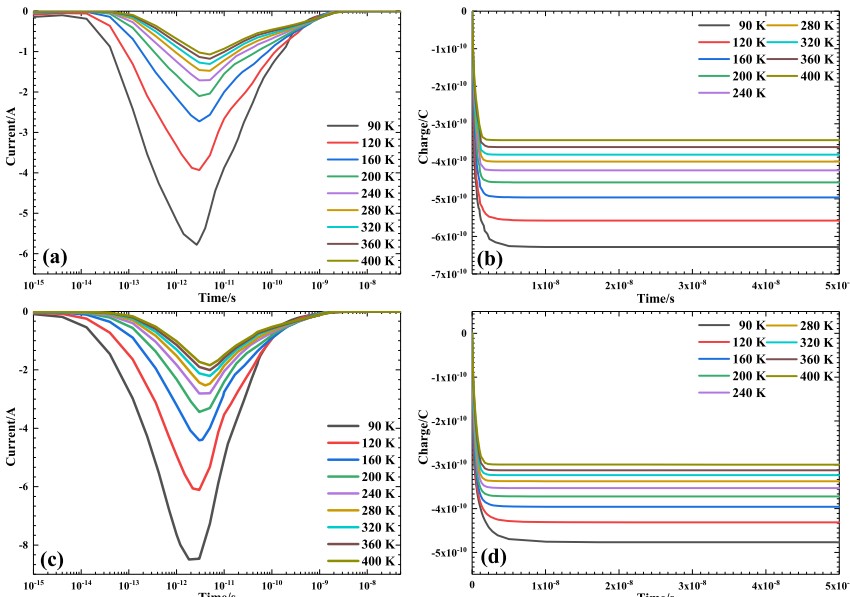

**Figure 12.** Change in base current and collected charge with time when heavy ions vertically irradiate the emitter (**a**,**b**) and base (**c**,**d**) of the SiGe HBT under different ambient temperatures.

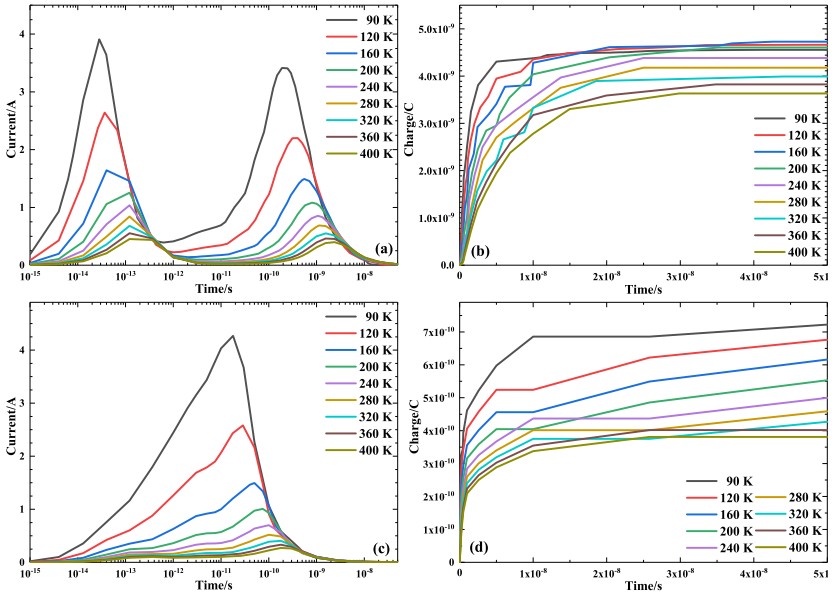

**Figure 13.** Change in collector current and collected charge with time when heavy ions vertically irradiate the collector (**a**,**b**) and substrate (**c**,**d**) of the SiGe HBT under different ambient temperatures.

### 3.6. Bias State

Previous studies have shown that the inverse bias of the large area C/S junction enhances the funnel effect, making the SiGe HBT sensitive to the single event effects. To compare the effects of different bias states on the SiGe HBT single event effects and considering the practical application in circuits, the positive bias (base = +1.2 V, collector = +3 V), off bias (emitter = +3 V, collector = +3 V), collector positive bias(collector = +3 V) and substrate inverse bias (substrate = −3 V), four kinds of work bias states that form the inverse bias C/S junction, were selected in this work. Figure 14 shows the current change in each terminal with time when heavy ions vertically irradiate the emitter of the SiGe

HBT under different bias states. When the SiGe HBT is in the substrate inverse bias state, electrons are collected by the collector and emitter, and holes are collected by the base and substrate. When the SiGe HBT is in the positive bias state, electrons are collected by the collector and base, and holes are collected by the emitter. When the SiGe HBT is in the off bias state, electrons are collected by the collector and emitter, and holes are collected by the base. When the SiGe HBT is in the substrate inverse bias state, electrons are collected by the collector and substrate, and holes are collected by the base and emitter. The current transient peak value when the SiGe HBT is in the collector positive bias state is the highest, followed by the SiGe HBT in the off bias and positive bias states, and the SiGe HBT in the substrate inverse bias state has the lowest current transient peak value. Therefore, the single event effect of the SiGe HBT is changed by the bias state during the heavy ions irradiation.

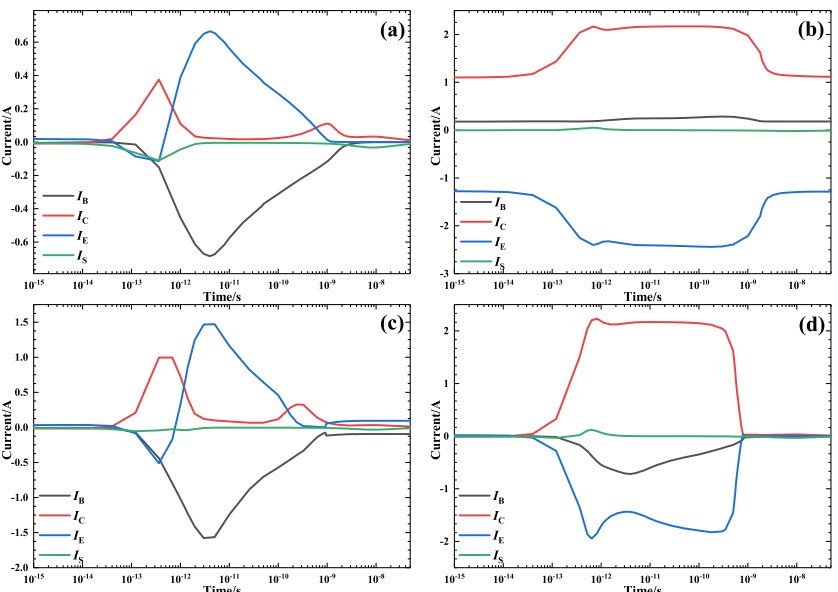

**Figure 14.** Change in current with time when heavy ions vertically irradiate the emitter of the SiGe HBT under the substrate inverse bias (**a**), positive bias (**b**), off bias (**c**) and collector positive bias (**d**) states.

Figure 15 shows the collected charge quantity change in each terminal with time when the heavy ions vertically irradiate the emitter of the SiGe HBT under different bias states. When the SiGe HBT is in the substrate inverse bias state, the collected charge quantity of the collector and substrate increases with time, while the collected charge quantity of the base and emitter rapidly reaches saturation after heavy ions irradiation, and the charge quantity of the electrons collected at the collector and emitter is equal to the charge quantity of the holes collected at the base and substrate. When the SiGe HBT is in the positive bias state, the collected charge of the base, collector and emitter increases with time, while the collected charge quantity of the substrate is not changed with time, and the charge quantity of the electrons collected at the base and collector is equal to the charge quantity of the holes collected at the emitter. When the SiGe HBT is in the off bias state, the collected charge quantity of the base, collector, emitter and substrate increases with time, and the charge quantity of the electrons collected at the collector and emitter is equal to the charge quantity of the holes collected at the base and substrate. When the SiGe HBT is in the collector positive bias state, the collected charge quantity of the collector and substrate increases slowly with time, while the collected charge quantity of the base and emitter rapidly reaches saturation, and the charge quantity of the electrons collected at the collector is equal to the charge quantity of the holes collected at the base, emitter and substrate. In the $5 \times 10^{-8}$ s after the heavy ions irradiation, the collected charge quantity of the terminal of the SiGe HBT in the positive bias state is the highest, followed by the SiGe HBT in the off

bias and collector positive bias states, and the least quantity of charge is collected at the terminal of the SiGe HBT in the substrate inverse bias state. Therefore, the quantity of the collected charge at each terminal of the SiGe HBT after the heavy ions irradiation depends on the bias state of the device.

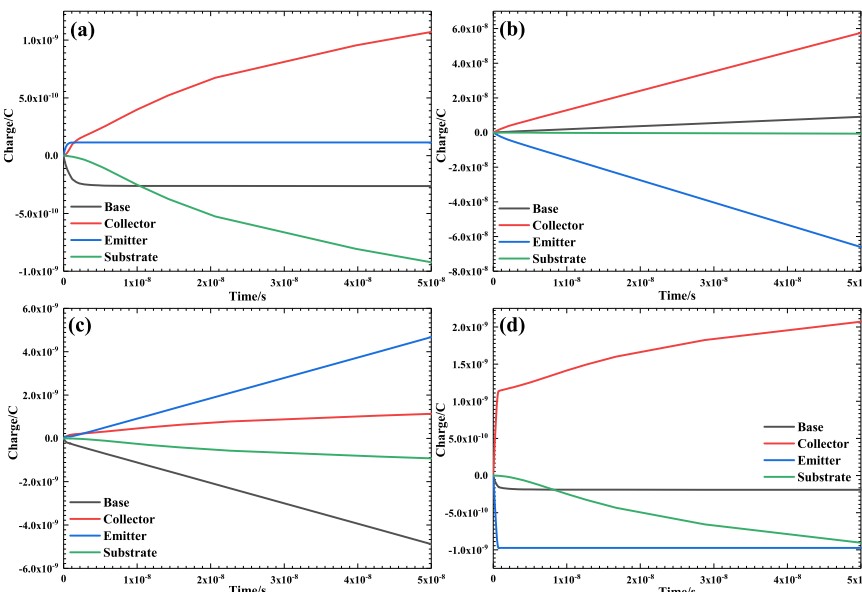

**Figure 15.** Change in collected charge with time when heavy ions vertically irradiate the emitter of the SiGe HBT under the substrate inverse bias (**a**), positive bias (**b**), off bias (**c**) and collector positive bias (**d**) states.

Figures A1–A3 show the change in the current with time when the heavy ions irradiate the base, collector and substrate of the SiGe HBT under the different bias states, respectively. When the heavy ions irradiate the base, the electrons are collected by the collector and emitter, and the holes are collected by the base and substrate under the substrate inverse bias and off bias. The collector and base collect the electrons under the positive bias state, while the emitter collects holes. Under the collector positive bias state, the electrons are collected by the collector and substrate, while the holes are collected by the base and emitter. The bias state not only changes the peak value and pulse shape of the current transients at each terminal but also changes the type of carriers collected at each terminal. When heavy ions irradiate the collector and substrate of the SiGe HBT, there are two peaks in the current of the collector and substrate, except for the positive bias state, and the peak values are not changed significantly by the bias states.

Figures A4–A6 show the collected charge quantity of each terminal when the heavy ions irradiate the base, collector and substrate under the positive bias, off bias and collector positive bias states, respectively. When the heavy ions irradiate the different locations of the SiGe HBT, the quantity of the collected charge at each terminal is changed by the bias state. The collected charge quantity increases linearly with time under the positive bias state and slowly with time under the other bias states. Therefore, the collected charge quantity at each terminal of the SiGe HBT is affected by both the irradiation position of the heavy ions and the bias state during the irradiation.

## 4. Conclusions

To understand the microphysical mechanism of single event effects in the SiGe HBT induced by heavy ion irradiation, the effects of the heavy ion striking location, incident angle, LET value, projected range, ambient temperature and bias state on the single event effects were investigated in this paper by using a TCAD simulation. The results show that the current transient peak value and collected carrier type of each terminal was changed by

these factors. The current transient peak value increases with the LET and projected range of the heavy ions and decreases with the ambient temperature. The single event effects of the SiGe HBT are not only affected by the heavy ion irradiation parameters such as the incident angle, LET value and projected range, but they are also affected by the striking location, ambient temperature and bias state. The peak value of the current transient peak value increases with the projected range when the emitter, collector and substrate of the SiGe HBT are irradiated by heavy ions with the same LET value, while the current transient peak value decreases slowly with the projected range when the heavy ions irradiate the base, which indicates that the species of heavy ions should be taken into account when carrying out research on the single event effects of the SiGe HBT induced by heavy ions irradiation. The main reason for the change in the single event effects is the change in the carrier mobility and transport mode under different factors. According to the simulation results, we can conclude that the single-particle effect caused by heavy ion irradiation can be weakened by increasing the pseudo electrode to carry away the electron–hole pairs generated by heavy ions irradiation, increasing the isolation area of the insulating materials to prevent the electron–hole pairs being collected by the terminals, increasing the ambient temperature to reduce the mobility of carriers, and other methods.

**Author Contributions:** Conceptualization, Z.Z. and G.G.; methodology, H.S. and J.L.; validation, Q.C.; data analysis, F.L. and S.Z.; resources, G.G. and X.O.; writing—original draft preparation, Z.Z.; writing—review and editing, Z.Z. and Q.C. All authors have read and agreed to the published version of the manuscript.

**Funding:** This research received no external funding.

**Data Availability Statement:** The data used to support the findings of this study are available from the corresponding author upon request.

**Conflicts of Interest:** The authors declare no conflicts of interest.

## Abbreviations

The following abbreviations are used in this manuscript:

| | |
|---|---|
| TCAD | Technology Computer-Aided Design |
| HBT | Heterojunction Bipolar Transistor |
| LET | Linear Energy Transfer |

## Appendix A

The following figures show the change in current and collected charge of each terminal with time when heavy ions irradiate the base, collector and substrate of the SiGe HBT under the different bias states, respectively.

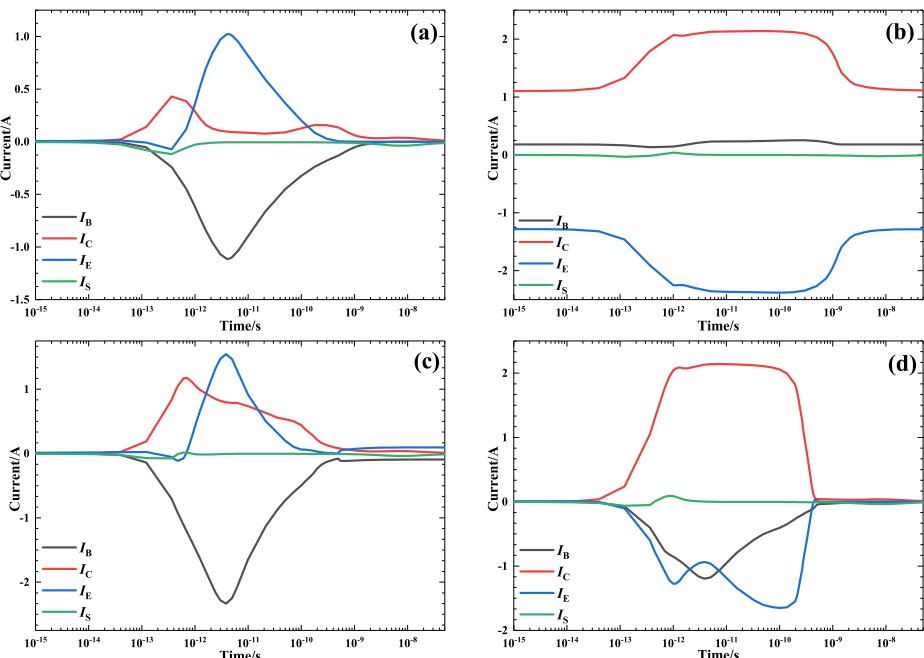

**Figure A1.** Change in current with time when heavy ions vertically irradiate the base of the SiGe HBT under the substrate inverse bias (**a**), positive bias (**b**), off bias (**c**) and collector positive bias (**d**) states.

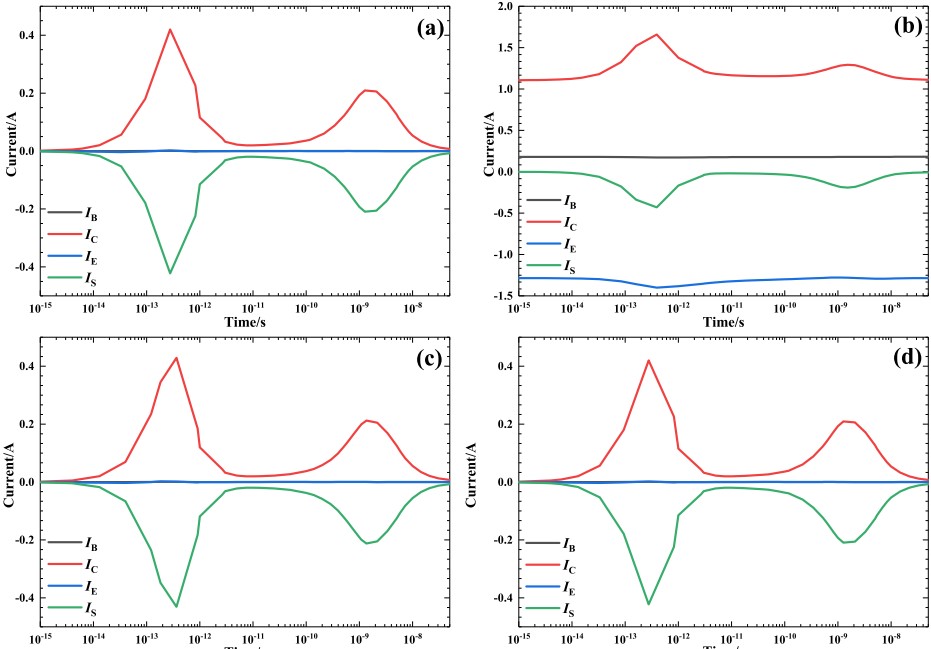

**Figure A2.** Change in current with time when heavy ions vertically irradiate the collector of the SiGe HBT under the substrate inverse bias (**a**), positive bias (**b**), off bias (**c**) and collector positive bias (**d**) states.

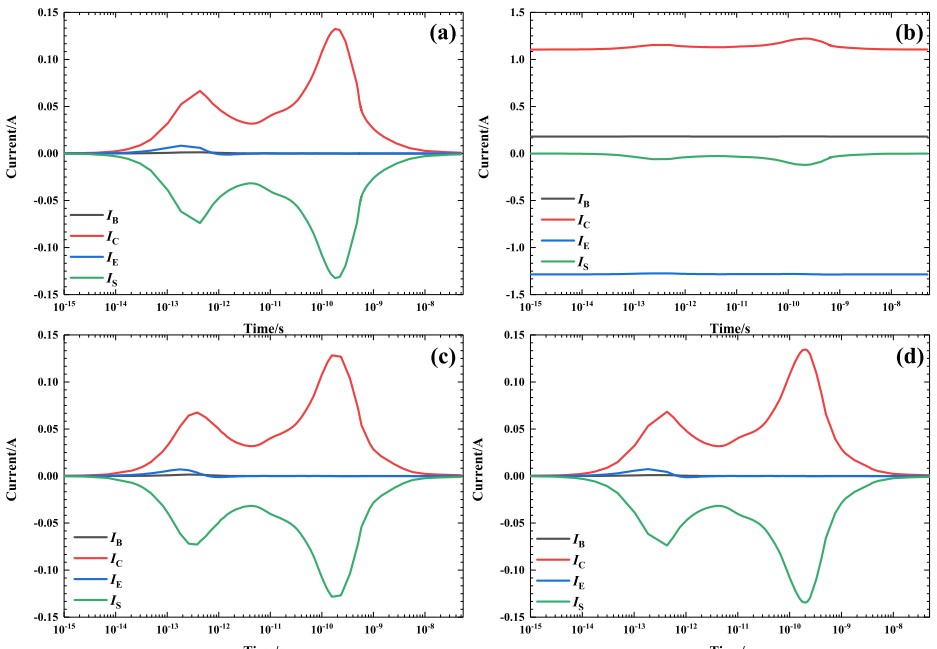

**Figure A3.** Change in current with time when heavy ions vertically irradiate the substrate of the SiGe HBT under the substrate inverse bias (**a**), positive bias (**b**), off bias (**c**) and collector positive bias (**d**) states.

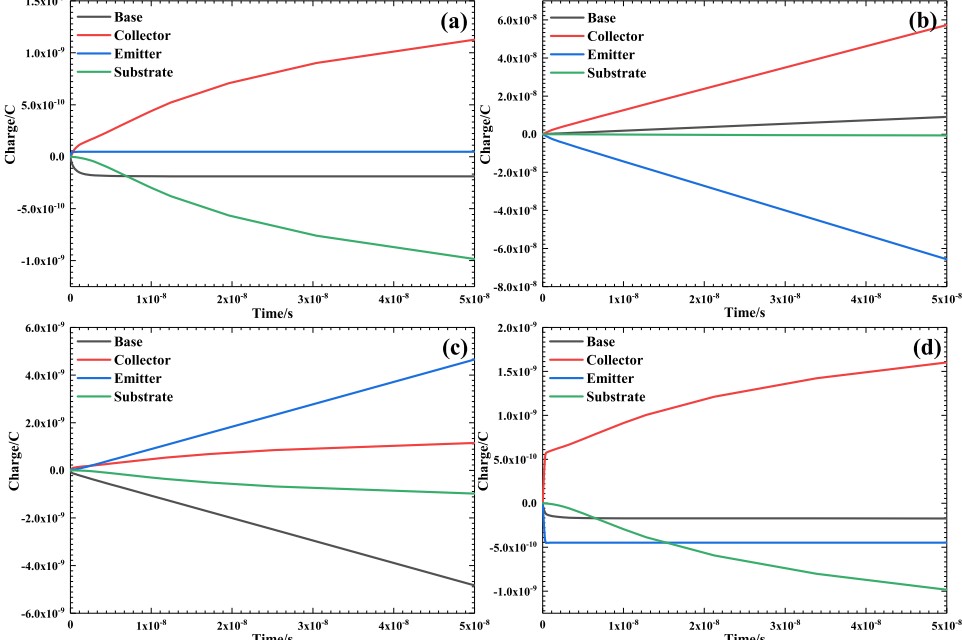

**Figure A4.** Change in collected charge with time when heavy ions vertically irradiate the emitter of the SiGe HBT under the substrate inverse bias (**a**), positive bias (**b**), off bias (**c**) and collector positive bias (**d**) states.

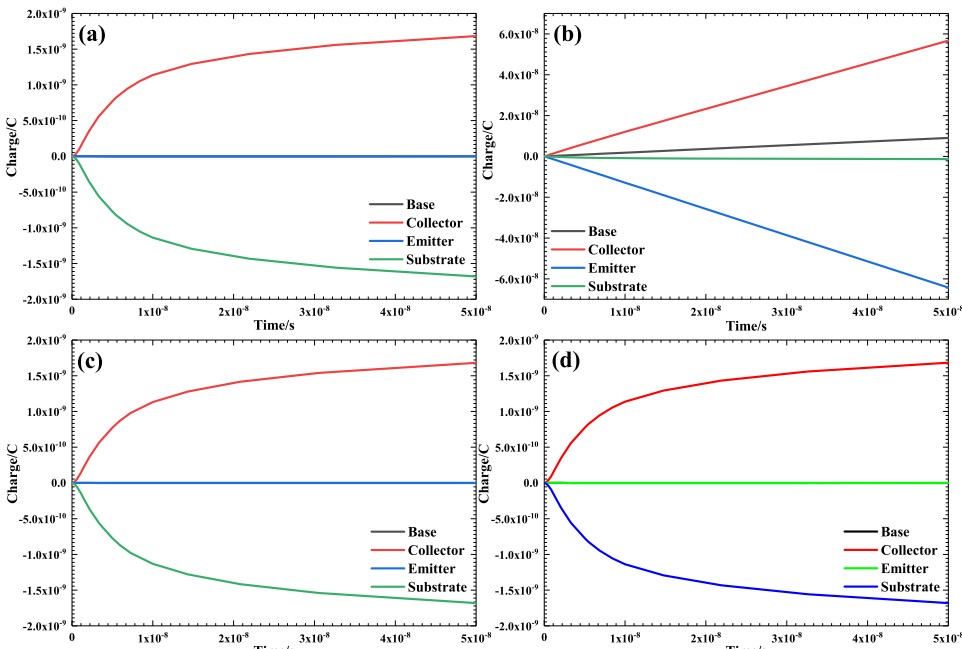

**Figure A5.** Change in collected charge with time when heavy ions vertically irradiate the emitter of the SiGe HBT under the substrate inverse bias (**a**), positive bias (**b**), off bias (**c**) and collector positive bias (**d**) states.

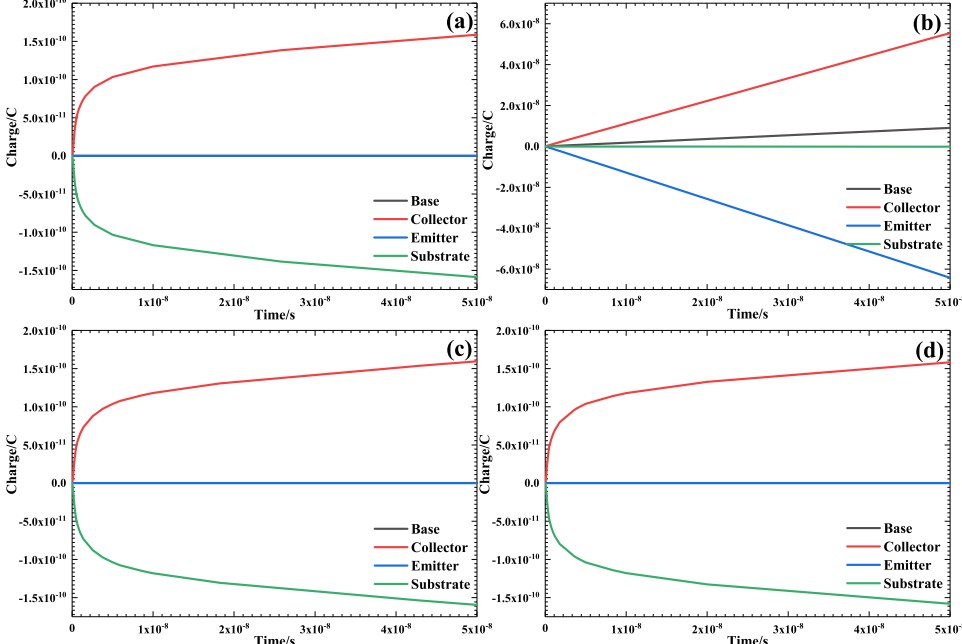

**Figure A6.** Change in collected charge with time when heavy ions vertically irradiate the emitter of the SiGe HBT under the substrate inverse bias (**a**), positive bias (**b**), off bias (**c**) and collector positive bias (**d**) states.

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
