# Peer review of "Effects of Different Factors on Single Event Effects Introduced by Heavy Ions in SiGe Heterojunction Bipolar Transistor: A TCAD Simulation"

_electronics, doi:10.3390/electronics12041008_

Round 1

Reviewer 1 Report

The English language in the draft is not incomprehensible, but it is difficult to read. I could not understand some sentences at all, due to syntax and word use. One example: "lighter ions will have longer project range". I don't know what "project range" means. Is it "projected range" perhaps? The term is used a lot, so I'm really missing something. Another example: what does "preparation material" mean in line 231? Regarding English syntax, most sentences in the draft needs to be revised to correct the grammar.  

The draft documents a lot of work done running TCAD simulations and presents a lot of observations from those simulations in words and in many plots. All of that is fine, if the language can be cleaned up. Less clear to me is what is the objective of a lot of it. Some statements made in the course of the presentation present nuggets of wisdom that no doubt accord with the observations from the simulation but are too qualitative and a-priori obvious. For example: line 240, "Therefore, the species of heavy ions should be considered when studying the single event effects of the SiGE HBT induced by heavy ions irradiation." I think anybody considering radiation effects on electronics would find that statement to be obviously true without doing any simulation.

In the end I find the conclusions to be supported by the results but to be too weak. In essence the conclusions seem to tell me little more than the observation that the various parameters studied do impact single-event effects. That is a generality that is to be expected, I think. Can the research instead tell me perhaps about what parameters are the most important to consider when trying to reduce single-event effects? Can they say something about how design and layout, or shielding, might mitigate the problem of single-event effects? Maybe they could compare two different technologies, or perhaps give me some idea about what minimum LET is necessary in order to have SEU problems with this technology. I'm not an expert in this, so probably there are better ideas. I just think that the conclusions should provide some useful information. Otherwise the paper will be useful only as a compendium of plots from which an expert reader might be able to draw their own conclusions.

I also note that the fonts in the plots are too small, rendering them difficult to read. That should be easy to remedy.

Some statements should be more precise. For example, in line 25, "a certain percentage of spacecrafts" means nothing. At least roughly say what percentage. In line 84 "its working period is shortened to a time scale of ps or several ns".  From ps to several ns is a range of 10,000 or more, so I'm left wondering what time scale is really being talked about.

Reviewer 2 Report

Effects of different factors on single event effects introduced by heavy ions in SiGe heterojunction bipolar transistor: A TCAD simulation

Review

            The impact that ion species, striking location, striking angle, projected range, temperature, bias state, and stopping power were correlated with single events in the SiGe HBT. The authors found that the transient peak correlated directly with dE/dx and the range of heavy ions and temperature. They also found correlations between transient peaks and striking angle and location. The implications for the finding are that single event peak transient research should be aware of the ion species of choice when performing either theoretical and experimental research in the field. The paper does a nice job of compiling the effects that single effects have on the peak transient values. I recommend publishing after more detailing of the simulations and extensive revision of the English grammar and syntax.

Detailed Comments

·      Section 3.4: Do you mean “projected range” instead “project range”?

·      You didn’t define “TCAD.” What software was used for the tcad simulation?

Additional Minor Comments

o   Line 15: “Due to the fact that the space” is what you mean in the first sentence.

§  This is an example of revisions that are needed on English syntax and grammar throughout the manuscript.

o   The labeling on all figures is rather small and difficult to read. Consider sizing up in all cases.

Round 2

Reviewer 1 Report

The authors have made substantial improvements according with my criticism. I think it can be published with some moderate editing of the language by the journal. 

Reviewer 2 Report

The revisions are acceptable.